# The Agri-Environment-Climate Measure as an Element of the Bioeconomy in Poland—A Spatial Study

Aleksandra Jezierska-Thöle [1,2,*], Roman Rudnicki [3], Łukasz Wiśniewski [3], Marta Gwiaździńska-Goraj [4] and Mirosław Biczkowski [3]

1. Institute of Geography, Kazimierz Wielki University, Plac Kościeleckich 8, 85-033 Bydgoszcz, Poland
2. Department of Anthropogeography, Freie University Berlin, Malteserstr. 74-100, 12 249 Berlin, Germany
3. Department of Spatial Planning and Tourism, Nicolaus Copernicus University in Toruń, Lwowska 1, 87-100 Toruń, Poland; rudnicki.r@umk.pl (R.R.); lukaszwisniewski@umk.pl (Ł.W.); mirbicz@umk.pl (M.B.)
4. Department of Socio-Economic Geography, University of Warmia and Mazury in Olsztyn, Prawochenskiego 15, 10-720 Olsztyn, Poland; marta.gwiazdzinska-goraj@uwm.edu.pl
* Correspondence: alekjez@ukw.edu.pl; Tel.: +48-52-327-71-45

**Abstract:** The Polish agricultural economy has a chance to dynamically develop and influence the innovation policy in the EU model of bioeconomy. The research aims to assess the spatial diversification of the level and structure of spending funds for two Rural Development Program (RDP) measures: agri-environment-climate measures (AECM) and organic farming scheme (OFS) aimed at supporting proenvironmental forms of agricultural management in the context of bioeconomy development. The EU financial perspective determined the time range for 2014–2020. The study was conducted on the example of Poland in two spatial scales: regional (province) and local (community). The analysis was based on partial indicators, which were then subjected to the standardisation procedure and included in the total as a synthetic indicator of the utilisation of RDP 2014–2020 funds aimed at supporting proenvironmental forms of farming. The following information was included in the evaluation: the number of farms, the size of utilised agricultural area (UAA) covered by support and the amounts of payments made under the two analysed RDP measures. In the research, the size and distribution of farms benefiting from AECM and OFS were determined. Besides, the relationship between funds absorption and socioeconomic development, as well as natural and non-natural conditions, were identified. The synthetic indicator of AECM/OFS usage showed a strong spatial differentiation, determined by the impact of several conditions: the level of socioeconomic development, the level of agriculture development, natural conditions of agriculture, land with significant natural and ecological values, and proenvironmental forms of land use on farms. Spatial diversification is more often the result of the impact of proenvironmental or natural-ecological factors than of socioeconomic conditions, or the level of agricultural development.

**Keywords:** bioproduction; CAP payments; sustainable agriculture; Poland

## 1. Introduction

Poland ranks third in Europe after France and Spain in terms of the share of agricultural land in the total area of the country (56%). In 2018, the global production value (in current prices) of Polish agricultural holdings ranked the country's agriculture 7th in the European Union, behind France, Germany, Italy, Spain, the UK, and the Netherlands [1,2]. Following the definition of the European Commission (EC) bioeconomy is "one of the oldest economic sectors known to humanity, and the life sciences and biotechnology are transforming it into one of the newest" [3,4]. Bioeconomy, i.e., industry based on bio-based raw materials and biotechnology, is concentrated around traditional sectors: agriculture, forestry, and food processing [5–7]. Polish agriculture may become an essential element in the development of the bioeconomy by supplying critical resources [8–10]. Bioeconomy is an important branch of the Polish economy, responsible for about 20% of the employment

and 10% of the total production volume [11]. The development of bioeconomy is determined by the depletion of available natural resources, climate change and the need to implement sustainable agriculture [12–14]. The key determinants of bioeconomy development are the adopted legal regulations implementing international obligations in the form of the UN sustainable development goals and climate and energy policy combined with innovation [15–17]. The Polish bioeconomy was formed under the commitments resulting from the membership in the European Union, the communication of the European Commission to the European Parliament and the Council of Europe titled "Innovation in the service of sustainable growth: bioeconomy for Europe" contributed to the development of bioeconomy in Poland [8,18,19]. References to bioeconomy are found in documents such as Strategy for Innovation and Efficiency of the Economy [20], Strategy for Sustainable Rural Development [21], and the Energy Security and Environment Strategy, all of which promote growing efficiency of the use of natural resources and raw materials [22]. The Organization for Economic Cooperation and Development (OECD) has produced a "policy agenda" pushing for biotechnology as a new "bioeconomy" [3,23–26].

Traditional bioeconomy includes primary production, i.e., agriculture, the development of which in Poland follows two tracks. The areas with a favourable agrarian structure are dominated by intensive agriculture with high rates of plant and animal production [23,27,28]. At the same time, it is accompanied by extensive, traditional agriculture and organic farming predominantly located in naturally valuable areas [29]. Bearing in mind that agricultural production is based on natural resources and its durability depends on the state of the natural environment, the type of agricultural production is of great importance (including industrial and bioenergy types), not only due to the quantity and quality of production, but also its impact on the natural environment and climate [30,31]. Bioeconomy, based on biodiversity, is of particular importance in agricultural areas which are protected and financially supported by EU programs [32]. One of the challenges for agriculture is to ensure food security while maintaining the postulates contained in the concept of bioeconomy [8]. Agricultural production has a significant impact on the natural environment, including responsibility for a significant part of greenhouse gas emissions [33] and at the same time is a sector susceptible to climate change [34]. These relationships are two-sided: environmental resources determine the size and directions of agricultural production; at the same time, agriculture changes the existing ecosystems, shapes the landscape and affects the individual components of nature [35–37].

Increased competition on the market and pressure to increase the agricultural production efficiency in the EU contribute to the loss of biodiversity, the disappearance of traditional forms of farming and local varieties of crop plants [38]. Therefore, it becomes essential to reconcile the increase in agricultural productivity and its competitiveness with the simultaneous reduction of its negative impact on the environment. Therefore, agricultural production must use energy, water and soil in a more effective and proenvironmental way, while reducing greenhouse gas emissions [19,39–41].

The answer to the above challenge is more sustainable agriculture, which combines production, economic, social, and ethical priorities with ecological safety [42,43]. The concept of sustainable development postulates the simultaneous implementation of goals relating to three independent but related areas: environmental (ecological), social and economic [44,45]. Sustainable agriculture could become an essential element in the development of the bioeconomy [46–49]. According to Kłodziński [50], it should be remembered that sustainable development of rural areas requires, above all, a compromise between agricultural producers, whose aim is to maximise the effects of their activities, and the interests of society, for which protection and management taking into account the state of the natural environment is becoming more and more critical. In the conditions of the Common Agricultural Policy (CAP), this leads to a redefinition of the concept of agriculture: from a narrow, productive approach, to holistic, sustainable and rational management of natural resources recognised as particularly protected public goods [51].

From the beginning of Poland's membership in the EU, i.e., 2004, the most effective CAP program aimed at minimising the negative impact of agriculture on the environment was the Agro-environmental Program. It had three goals: protection of the environment and landscape, development of organic farming and preservation of biodiversity. Implemented in the first (incomplete) financial period (2004–2006), it continued in 2007–2013, and now functions under the Rural Development Program (RDP, 2014–2020 perspective). In the concept of multifunctional and sustainable agricultural development adopted for implementation, two measures are of particular importance for shaping the relationship between agriculture and the environment—agri-environment-climate measures (AECM) and organic farming scheme (OFS; in previous EU financial periods, these measures were included in one agri-environmental program). It should be emphasised that agri-environment-climate measures of the RDP are one of the financial instruments of the bioeconomy and are part of its development trend [52–54]. These activities are mainly aimed at strengthening two nonmarket functions of agriculture:

- Green—related to the management of land resources in order to maintain their valuable properties, with the creation of conditions for wild animals and plants, protection of animal welfare, maintenance of biodiversity and improvement of the circulation of chemicals in agricultural production systems [11,55];
- blue—related to water resources management, water quality improvement, flood prevention, hydropower and wind energy generation [5].

When considering the multifunctionality of agriculture, support for farms in areas with unfavourable farming conditions is aimed at securing their possibility of further operation. Land management and land use should be based on environmentally friendly principles, while supporting functions other than food production, thus preventing trends of marginalisation and degradation of these areas [43,56]. The answer to the problems related to sustainable development is to improve the methods of managing the environment and natural resources. In this context, social-ecologically sustainable agricultural management is gaining more and more importance [40,57].

The European Bioeconomy Strategy developed by the EC (adopted on 13 February 2012) is based on three pillars [58], that is the support from EU and national funds, providing knowledge for sustainable production growth and the creation of a bioeconomy panel and bioeconomy observatory. Bearing the above in mind, the article presents and describes the first pillar of the strategy, i.e., supporting the bioeconomy with EU funds from the RDP [59].

The main objective of the research is to assess the spatial diversity of farms acquiring CAP funds aimed at supporting sustainable agriculture, i.e., concerning two RDP measures 2014–2020, "Agri-Environment-Climate Measures" and "Organic Farming Scheme". The second aim of the study is an attempt to delimit the determinants affecting the level of use of the researched funds, and thus the possibilities of sustainable agricultural development.It was assumed that the identification of such targeted activity of farmers is a sine qua non condition for broader inclusion of agriculture in the framework of bioeconomy [60].The research goals set in this way will help to answer the question "Can the spatial diversification of the use of proecological CAP funds be the key to a more regionally and locally optimised development of the bioeconomy in Polish agriculture?"

The research used an indicator of the share of completed applications of the measures mentioned above in the total number of farms to assess the level of interest of farmers towards proenvironmental forms of agricultural management along with its spatial diversification. Additionally, the strength of the relationship (correlation) between the level of activity determined in this way and the adopted conditions thus identified i.e., the levels of socioeconomic and agricultural development, and two environmental determinants related to the assessment of the natural conditions of agriculture and the share of proenvironmental forms of land use.

Earlier studies [61–65] indicate territorial disparities in obtaining CAP funds, resulting from the characteristics of a given area, both human (socioeconomic) and environmental.

## 2. Materials and Methods

### 2.1. Study Area and Materials

Taking up the above topic was motivated by the need to summarise the effects of two RDP measures of 2014–2020 AECM and OFS, which taken together constituted the basis for recognising the strength of the relationship between the different level of absorption of funds from the measures as mentioned earlier and the level of natural and non-natural conditions. The spatial scope of the research covered the territory of Poland (NUTS 0) in the system of province (16 NUTS 2 units) and communities (2477 units; the third-order administrative division of the country sometimes referred to as "communes" or "municipalities", until 2016—according to Local Administrative Units—LAU level 2).

Data on the implementation of AECM/OFS were obtained from the Agency for Restructuring and Modernisation of Agriculture (ARMA; Warsaw, Poland). They took into account the number of beneficiaries, the area covered by the payment and the amounts of payments made. The second leading source of data was the Local Data Bank of the Central Statistical Office (Warsaw, Poland) [66–68]. The obtained data from the ARiMR related to two RDP 2014–2020 measures (AECM, OFS), including:

- the number of completed applications—97,200, which constituted 10.4% of the total number of farms;
- the surface of the subsidised area—1,259,600 ha, which constituted 10.8% of the total agricultural area of farms;
- the realised payments—EUR 933.8 million (at the rate of PLN 4.295 to EUR 1), which was EUR 1190 per farm.

The data concerned a wide range of issues that allowed for spatial assessment, among others, of the level of socioeconomic development, the level of agricultural development or natural and ecological valorisation, which were adopted as the level of conditions for the sustainable development of bioagriculture (see Table 1).

The main criteria for assessing spatial differentiation were the number of farms, the area of UAA covered by the support and the amounts of AECM/OFS payments made. The research assumptions included analysis in two spatial scales:

- macroscale—comprehensive nationwide analysis;
- microscale—enabling the identification of specific areas in which activities aroused extreme interest and areas in which farmers showed passivity in applying for funds for agri-environment-climate activities. Such an approach is an advantage of the work, as most of the analyses related to the evaluation of the implementation of EU funds are conducted only on a regional scale, without in-depth analysis at the local level (LAU 2 units).

The primary analysis was based on the number of applications completed within the framework of the said measures and the volume of funds obtained. Both elements allow assessing the scale of farmers' interest in activities aimed at diversifying the sources of income. The empirical nature of the article, to a large extent, contributes to the development of the cognitive thread in the field of the impact of EU funds on the diversification of farmers' income sources and the development of entrepreneurship in rural areas towards the development of nonagricultural activities, with particular emphasis on the bioeconomy.

### 2.2. Methods

The implementation of the set research goal required the adoption of an appropriate research procedure and the construction of a whole set of indicators. The research was conducted in several stages (cf. Figure 1). In the first one, three partial indicators (IAF, ITR, IFSF) were used to assess the spatial level of the use of proenvironmental CAP funds (IUF-RDP). ARMA data and normalisation methods were used. The aim of the second stage was spatial delimitation of selected determinants (LSED, LAO, APS, NEA), which should determine the scale and directions of using the researched funds (IUF-RDP). They were defined on the basis of 12 partial indices. The last stage was a comparative analysis of

both planes, which allowed to assess the role of individual determinants in the level of the use of proenvironmental CAP funds.

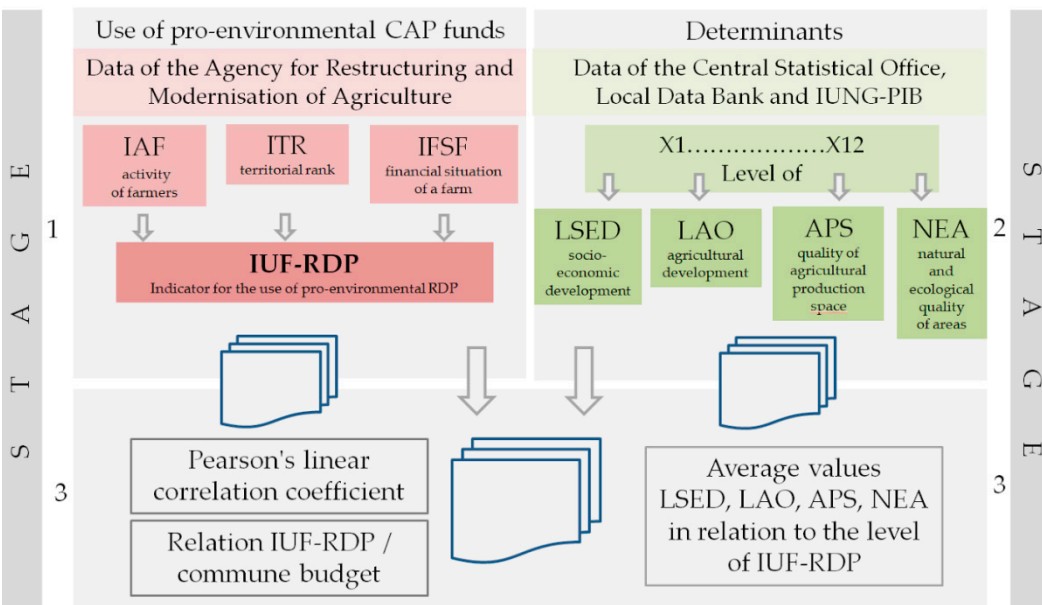

**Figure 1.** Research procedure.

2.2.1. Stage 1. Indicator of the Use of Proenvironmental CAP Funds

The basis for assessing the use of proenvironmental CAP measures were normalised values of three indicators illustrating:

- (IAF) the activity of farmers in terms of obtaining funds (ratio of the number of applications to the total number of farms expressed in percentages);
- (ITR) territorial rank (the ratio of the number of applications to the area of agricultural land expressed in percentages);
- (IFSF) impact on the financial situation of a farm (ratio of obtained subsidies in EUR per farm). The above indicators were subjected to the normalisation procedure following the formula [69–71].

$$Z_{ij} = \frac{X_{ij} - av.X_i}{\delta_i} \tag{1}$$

where:

$Z_{ij}$—standardised value of the diagnostic feature '$i$' in spatial unit '$j$'

$X_{ij}$—value of diagnostic feature '$i$' in spatial unit '$j$'

$av.X_i$—average value of diagnostic feature '$i$'

$\delta_i$—standard deviation of diagnostic feature '$i$'.

The next step was to calculate the synthetic indicator for the use of proenvironmental RDP funds (IUF-RDP), according to the formula:

$$G = \frac{1}{M}\big(Z_{i1} + Z_{i2} + \ldots + Z_{ij}\big) \tag{2}$$

where:

$G$—average standardised value of selected diagnostic features within the respective group of features

$Z_{ij}$—standardised value of diagnostic feature '$i$' in spatial unit '$j$'

$M$—number of diagnostic features.

The zero values (national means) of the indicators, assuming a standard deviation threshold of $+/-0.5$, were the basis for distinguishing three classes characterised by low (below $-0.50\ \delta$), medium (from $-0.50$ to $0.50\ \delta$), and high (above $0.50\ \delta$) level of the phenomenon.

### 2.2.2. Stage 2. Assessment Planes—Determinants of Sustainable Development of (Bio)Agriculture

The spatial distribution of the level of the use of proenvironmental CAP funds was compared to synthetic indicators illustrating the level of socioeconomic development (LSED), the level of agricultural development (LAD), the quality of agricultural production space (APS), and the natural and ecological quality of areas (NEA). The indicators were constructed based on normalisation and then averaging of several partial indicators and division into units with the low, medium and high level of the studied phenomenon (the same as in the case of IUF-RDP). The basis for delimiting selected indicators is presented in Table 1.

**Table 1.** Assessment planes—determinants of development of sustainable agriculture in Poland characteristics. Source: own elaboration on [66–68,72,73].

| Assessment Plane | Delimitation Indicators | Data Source |
|---|---|---|
| The level of socioeconomic development (LSED) | ($x^1$) business entities in the REGON register per 10,000 population | [70] |
| | ($x^2$) unemployed per 10,000 population (destimulant) | [70] |
| | ($x^3$) population with the access to the sewage network as a percentage of the total population | [70] |
| | ($x^4$) own incomes of communities in PLN per capita | [70] |
| The level of agricultural development (LAD) | ($x^5$) the average area of a farm in ha (2018, according to ARMA) | [70] |
| | ($x^6$) farmers with secondary and higher education as a percentage of the total number of farmers | [66] |
| | ($x^7$) young farmers (up to 34 years of age) as a percentage of the total number of farms | [66] |
| | ($x^8$) noncereal crops as a percentage of the total sown area | [66] |
| The quality of agricultural production space (APS) | ($x^9$) indicator of the quality of agricultural production space | [67] |
| The natural and ecological quality of areas (NEA) | ($x^{10}$) forests and wooded and bushy land as a percentage of the total area | [70] |
| | ($x^{11}$) grasslands, water bodies, and the legally protected areas as a percentage of the total area (2018, according to CSO BDL) | [70] |
| | ($x^{12}$) priority zones of the agri-environmental program delimited in the period 2004–2006 as a percentage of the total area | [68] |

### 2.2.3. Stage 3. Assessment of IUF-RDP in the Context of Conditions

The assessment of IUF-RDP in the context of separate groups of conditions for sustainable agricultural development was carried out in two ways. Firstly, it will support the calculation of the average values of LSED, LAD, APS, NEA indicators for each category (low, medium, high) for the level of use of IUF-RDP funds. In the analysis aimed at determining the strength and direction of the relationship between the synthetic index of the use of proenvironmental RDP measures and the conditions of sustainable agricultural development, the linear Pearson correlation coefficient (according to the product-moment) was used. Correlation factor took numerical values from $(-1)$ to $(+1)$, where the value with zero indicates no statistical relationship. Additionally, in order to better illustrate the rank of the studied group of funds, an indicator of the ratio (IR) of the value of proenvironmental

EU funds to the size of commune budgets (EU payments as % of the commune budget in 2015–2019) was introduced.

## 3. Results and Discussion

The research questions and adopted methods mean that the discussion of the results is preceded by explanatory background. It is a theoretical framework on the essence and importance of agri-environment-climate action for the development of bioeconomy and implementation of the agri-environmental program in Poland in 2007–2013 against the European Union.

### 3.1. The Role and Importance of Agri-Environment-Climate Action

One of the objectives of the EU's Common Agricultural Policy is to promote environmentally friendly agricultural practices. This goal is being implemented, among others, by implementing agri-environment-climate measures in all EU member states. These measures are part of the Rural Development Program for 2014–2020 (RDP 2014–2020) and are mostly a continuation of the previous measure, RDP agri-environmental program for 2007–2013 [73]. It was planned as one of the components implementing strategic EU and national environmental goals, taking into account the economic and social importance of agriculture in the context of the growing demand for agricultural raw materials, including for the bioeconomy, and the still high importance of agricultural activity for employment and territorial development in Poland [74,75]. The agri-environmental program is a financial instrument aimed at encouraging farmers to continue or apply agricultural practices leading to the greening of agricultural production. The implementation of the agri-environmental program contributes to the sustainable development of rural areas and the preservation of biodiversity in these areas. The primary assumption of the program is to promote agricultural production based on methods consistent with the requirements of environmental and nature protection [76–78]. An additional goal of the program is to increase the environmental awareness of the rural community. According to research [47,79], the agri-environmental program has become an impulse mainly for the development of multifunctional agriculture.

In the period 2004–2006, the agri-environmental program included seven packages, while in the years 2007–2013, nine packages were implemented (RDP 2004–2006 and RDP 2007–2013). The agri-environment-climate measure (RDP 2014-2020) consists of seven packages: (1) Sustainable agriculture, (2) Soil and water protection, (3) Preservation of orchards with traditional varieties of fruit trees, (4) Valuable habitats and endangered bird species in Natura 2000 areas, (5) Valuable habitats outside Natura 2000 areas, (6) Conservation of endangered plant genetic resources in agriculture, and (7) Preservation of endangered animal genetic resources in agriculture.

The limit of funds for Poland for 2014–2020 for this priority was approximately EUR 4.2 billion [80] and addressed the implementation of the following specific objectives:

- restoring, protecting, and enhancing biodiversity, including in Natura 2000 sites and areas facing natural or other specific constraints, and HNV farming and the state of European landscapes;
- improving water management, including fertilisation and pesticide use,
- preventing soil erosion and improving soil management [81].

The limit of funds allocated to Poland for this measure for 2014–2020 is approximately EUR 1.4 billion, which constitutes 33% of Priority 4. These measures (excluding Package 2. Organic farming under the agri-environment-climate measures) were implemented for a total of 11.6% of the area reported for direct payments (as of 2015). In subsequent campaigns, this area was systematically decreasing, and in 2018 the estimated area amounted to 7.1% of the area reported for direct payments. In the context of specific Priority 4: Preventing soil erosion and improving soil management, the most crucial action is the organic farming scheme (OFS). The total funds allocated for this purpose amount to less than EUR 700 million, which constituted approximately 17% of Priority 4 funds. In total,

by the end of 2018, based on issued decisions granting ecological payment, the size of the physical area covered by support was 531,816 ha, which was 3.7% of the area reported for direct payments. Although this is a highly unsatisfactory effect, in the years 2016–2018, a reasonably stable increase (approximately 12% per year) of the agricultural area on which this measure is implemented was observed.

As part of agri-environmental programs, which are followed by specific financial support, farmers are encouraged to act to protect the natural environment, biodiversity, and preserve landscape values. The beneficiary may join the measure, first of all, if the farm has at least 1 ha of UAA (3 ha in Package 1. Sustainable agriculture) or at least 1 ha of natural areas and has the producer's identification number assigned by the Agency for Restructuring and Modernisation of Agriculture.

### 3.2. Implementation of Proenvironmental CAP Funds in Poland in 2014–2020

In Poland, in 2014–2020, 97,200 projects were implemented as AECM/OFS activities, including 67,200 for the operation of AECM and 30,100 of OFS. These ranged from less than 1000 in Opolskie (700) and Śląskie (900) to over 10,000 in Lubelskie (13,300), Podlaskie (11,700), and Warmińsko-Mazurskie (10,400; see Table 2, Figure 2a). The researched group of farms accounted for 10.4% of the total number of farms in Poland (IAF; according to the register of agricultural producers by the ARMA).

**Table 2.** Proenvironmental instruments of the Rural Development Program (RDP) 2014–2020 in Poland—together agri-environment-climate (AECM) and organic farming (OFS) measures—selected elements. Source: own elaboration on [66–68,72,73].

| | Specification | Number of Applications | | Subsidised Area | | Payments Made | | IUF-RDP ** |
| | | In Total Thousand Requests * | IAF ** | In Thousand ha * | ITR ** | In Million Euros | IFSF ** | |
|---|---|---|---|---|---|---|---|---|
| | POLAND | 97.2 | 10.4 | 1259.6 | 10.8 | 933.8 | 1.190 | 0.00 |
| I | Dolnośląskie | 5.1 | 10.0 | 77.6 | 9.2 | 64.7 | 1.259 | −0.03 |
| II | Kujawsko-Pomorskie | 3.8 | 6.3 | 68.7 | 6.7 | 37.2 | 623 | −0.26 |
| III | Lubelskie | 13.3 | 7.9 | 119.3 | 8.8 | 87.8 | 522 | −0.19 |
| IV | Lubuskie | 4.7 | 24.4 | 98.4 | 24.1 | 83.2 | 4.304 | 1.05 |
| V | Łódzkie | 2.6 | 2.2 | 20.4 | 2.2 | 12.9 | 112 | −0.52 |
| VI | Małopolskie | 3.8 | 3.3 | 18.7 | 3.8 | 17.1 | 150 | −0.46 |
| VII | Mazowieckie | 7.4 | 3.7 | 57.0 | 3.1 | 40.4 | 204 | −0.46 |
| VIII | Opolskie | 0.7 | 2.6 | 17.2 | 3.4 | 8.1 | 302 | −0.46 |
| IX | Podkarpackie | 9.8 | 8.8 | 64.1 | 11.9 | 66.9 | 600 | −0.10 |
| X | Podlaskie | 11.7 | 14.9 | 102.9 | 10.1 | 82.1 | 1.051 | 0.07 |
| XI | Pomorskie | 6.1 | 16.2 | 109.9 | 15.2 | 70.6 | 1.889 | 0.33 |
| XII | Śląskie | 0.9 | 2.1 | 9.9 | 3.0 | 6.5 | 159 | −0.50 |
| XIII | Świętokrzyskie | 4.1 | 5.0 | 26.0 | 5.3 | 18.3 | 223 | −0.38 |
| XIV | Warmińsko-Mazurskie | 10.4 | 24.6 | 195.2 | 20.1 | 145.8 | 3.460 | 0.85 |
| XV | Wielkopolskie | 5.3 | 4.5 | 76.2 | 4.4 | 48.7 | 419 | −0.38 |
| XVI | Zachodniopomorskie | 7.8 | 28.2 | 198.4 | 23.5 | 143.6 | 5.173 | 1.25 |

* annual average values 2015–2019, (for 2281 communities); ** IAF—the activity of farmers; ITR—territorial rank; IFSF—impact on the financial situation of a farm; IUF-RDP—indicator for the use of proenvironmental RDP (more about indicators: see methodology).

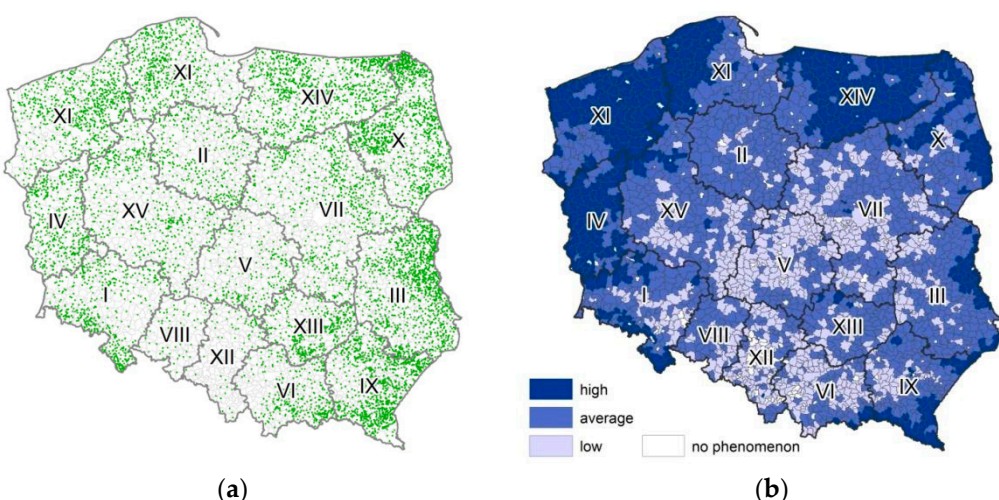

(a)    (b)

**Figure 2.** Number of submitted applications (**a**) and the synthetic indicator of the utilisation of the Rural Development Program (IUF-RDP-indicator for the use of proenvironmental RDP (**b**)). *1 dot = 10 applications. Source: own elaboration on [66–68,70,71].

In the province system, the most significant activity was recorded by Zachodniopomorskie (28.2%), Warmińsko-Mazurskie (24.6%), and Lubuskie (24.4%), the lowest by Śląskie (2.1%), Łódzkie (2.2%), and Opolskie (2.6%). The UAA covered by the cofinancing was 1,284,900 ha, while the ITR was 10.8%. The spatial analysis showed that the highest ITR was recorded in Lubuskie (24.1%) and zachodniopomorskie (23.5%), while the lowest in Łódzkie (2.2%) as well as Mazowieckie (3.1%) and Śląskie (3.0% each).

The AECM and OFS funds are commonly used in communities with natural and ecological values ($X^4$, $X^5$). The communities showing high interest in proenvironmental forms of EU support include Gołdap (534 applications), UstrzykiDolne (506), UjścieGorlickie (386), and DrawskoPomorskie (379). Generally, however, the group of the most active communities is relatively small. More than 100 applications were implemented only in 261 communities (10.5% of the total). The group of communities where the beneficiaries showed passivity and distance to the implementation of proecological activities is much more numerous. It is confirmed by this study, as it shows that in 338 communities, only 1–5 applications were implemented, while in another 304 communities, 6–10 applications. In total, these administrative units constitute as much as a quarter of all communities in Poland. Besides, there is a relatively large group of communities (198, i.e., 8%) in which none of the farmers took advantage of the possibility of obtaining EU payments for proecological activities. Thus, the activity rate of beneficiaries is relatively low. Statistically, the leaders among local authorities in implementing solutions supporting the bioeconomy are found in one in 10 communities in Poland.

Bioeconomy, based on biodiversity, is of particular importance in areas used for agriculture, which are covered by EU payments for proenvironmental measures, i.e., in this case from AECM and OFS. Hence, one of the essential measures in the spatial dimension is the UAA to which payments under the instruments above have been granted. This indicator shows extensive spatial spans. From this perspective, the leading communities in Poland include Gołdap (the area covered by AECM and OFS payments is 8885 ha), Szczecinek (8617), DrawskoPomorskie (7035), Słońsk (6388), Bobolice (6322), and BiałyBór (6290). In total, in 387 communities, more than 1000 ha of UAA were qualified for the subsidies. In the next 299 communities it was 500–999 ha, and in as many as 918 from 100 to 499 ha. In total, the area eligible for support for proecological activities in Poland, i.e., AECM and OFS, amounts to 1.259 million ha, i.e., approximately 10.8% of the total UAA. For comparison, in Germany—the most developed EU country in this respect—subsidies to the area covered by the agri-environmental program were granted to nearly 5.3 million ha, i.e., about one-fourth of the total UAA [81].



The total value of cofinancing was EUR 933.8 million (on average, EUR 145 per farm), the majority of which was related to the AECM (68.6%). In terms of the volume of the payments, it was shown that an average farm received almost EUR 1200 (IFSF). This type of impact on the financial situation of farms was the highest in Zachodniopomorskie (EUR 5173). At the community level, the threshold of EUR 10,000 was exceeded in 34 of them, while two communities (Stepnica in Zachodniopomorskie and Lutowiska in Podkarpackie) exceeded EUR 19,000.

It was assumed that IUF-RDP (average IAF, ITR, IFSF) is a determinant of proenvironmental preferences of farms, therefore recognising its spatial diversity is an essential element of bioeconomy development opportunities.

The synthetic indicator of the utilisation of the AECM and OFS resources (normalised average) showed high spatial differentiation. In terms of regions, the leader in the use of funds was Zachodniopomorskie (1.25), followed by Lubuskie (1.05) and Warmińsko-Mazurskie (0.85), which achieved the highest values in the system of partial indicators (i.e., activity and absorption). Low values of the AECM/OFS utilisation rate were noted in Łódzkie (−0.52) and Śląskie (−0.50). Such a territorially oriented implementation of AECM and OFS measures shows that the use of biodiversity in agriculture is pragmatic, which increases the possibilities for sustainable use and good management of environmental resources. On the other hand, it also helps to strengthen the environmental ecosystem, which will help preserve biodiversity at the farm level and beyond. Appropriate (i.e., in line with environmental preferences, as well as agricultural and economic opportunities) territorially oriented AECM and OFS activities should significantly strengthen the desired economic profile, i.e., aimed at the bioeconomy. However, it should be noted that the implementation of proecological activities by farmers is not only motivated by care for the environment. As such, no other behaviours are mainly due to the economic factor associated with the possibility of obtaining subsidies from the CAP funds. It is evidenced by the fact that the activities of AECM and OFS in Poland are used mainly by large farms located in the northern and western parts of the country (Zachodniopomorskie, Warmińsko-Mazurskie, Lubuskie). Thus, it points to the opposite relationship than, for example, in neighbouring Germany, where smaller farms, focused on more extensive forms of management, are more eager to use funds from proecological activities.

In the system of communities, the highest level of activity in the use of EU funds was found in Dziwnów (8.6), Zachodniopomorskie and GórowoIławeckie (7.90), Warmińsko-Mazurskie. The remaining clusters of communities are located in the coastal and lake district regions, within the Natura 2000 areas, within the territory of national parks, e.g., Słowiński (NP communities of Łeba 5.62, Smołdzino 5.2) and Bieszczady NP (community of Lutowiska 4.9) (cf. Figure 2b).

The value of IUF-RDP was the basis for the classification of communities into three groups, namely with the low, medium, and high levels of absorption of proenvironmental RDP funds (see Figure 2b).

### 3.3. An Attempt at Spatial Delimitation of Conditions for the Development of Sustainable Agriculture in Poland

Apart from recognising the spatial differentiation of the use of proenvironmental RDP funds, one of the objectives of the study was an attempt to delimit the spatial conditions influencing the sustainable development of agriculture, and thus the absorption of these funds. The analysis of the values of synthetic indicators (LSED, LAD, APS, NEA) both on a regional (see Table 3) and local scale (see Figures 3 and 4) shows that the potential factors determining the sustainable development of agriculture create an extremely complex system.

**Table 3.** Selected conditions for the development of sustainable agriculture in Poland (indicators in the form of a normalised value). Source: own elaboration on [66–68,72,73].

| | Province | The Level of Socioeconomic Development | The Level of Agricultural Development | The Quality of Agricultural Production Space | The Natural and Ecological Quality of Areas |
|---|---|---|---|---|---|
| | | LSED * | LAD * | APS * | NEA * |
| I | Dolnośląskie | 0.30 | 0.21 | 0.39 | −0.12 |
| II | Kujawsko-Pomorskie | −0.35 | 0.36 | 0.39 | −0.27 |
| III | Lubelskie | −0.66 | −0.04 | 0.97 | −0.27 |
| IV | Lubuskie | 0.00 | 0.24 | −0.54 | 0.20 |
| V | Łódzkie | −0.16 | −0.11 | −0.27 | −0.41 |
| VI | Małopolskie | −0.03 | −0.24 | 0.03 | 0.17 |
| VII | Mazowieckie | 0.51 | −0.04 | −0.59 | 0.13 |
| VIII | Opolskie | −0.11 | 0.17 | 0.72 | −0.52 |
| IX | Podkarpackie | −0.61 | −0.29 | 0.27 | 0.40 |
| X | Podlaskie | −0.43 | 0.17 | −1.09 | 0.28 |
| XI | Pomorskie | 0.30 | 0.21 | 0.44 | −0.05 |
| XII | Śląskie | 0.13 | −0.36 | 0.15 | −0.25 |
| XIII | Świętokrzyskie | −0.57 | −0.21 | −0.03 | 0.29 |
| XIV | Warmińsko-Mazurskie | −0.46 | 0.36 | 0.15 | 0.17 |
| XV | Wielkopolskie | 0.20 | 0.14 | −0.28 | −0.10 |
| XVI | Zachodniopomorskie | 0.20 | 0.71 | −0.02 | 0.13 |

* LSED—the level of socioeconomic development; LAD—the level of agricultural development; APS—the quality of agricultural production space; NEA—the natural and ecological quality of areas.

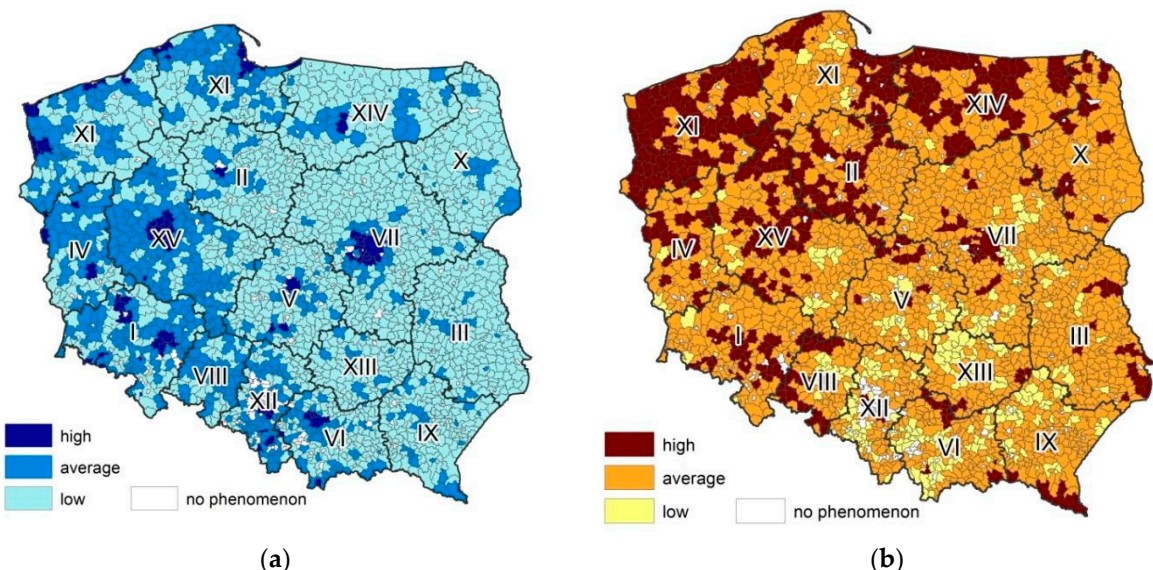

(**a**)                    (**b**)

**Figure 3.** The level of socioeconomic development ((**a**) LSED) and the level of agricultural development ((**b**) LAD). Source: own elaboration on [66–68,70,71].

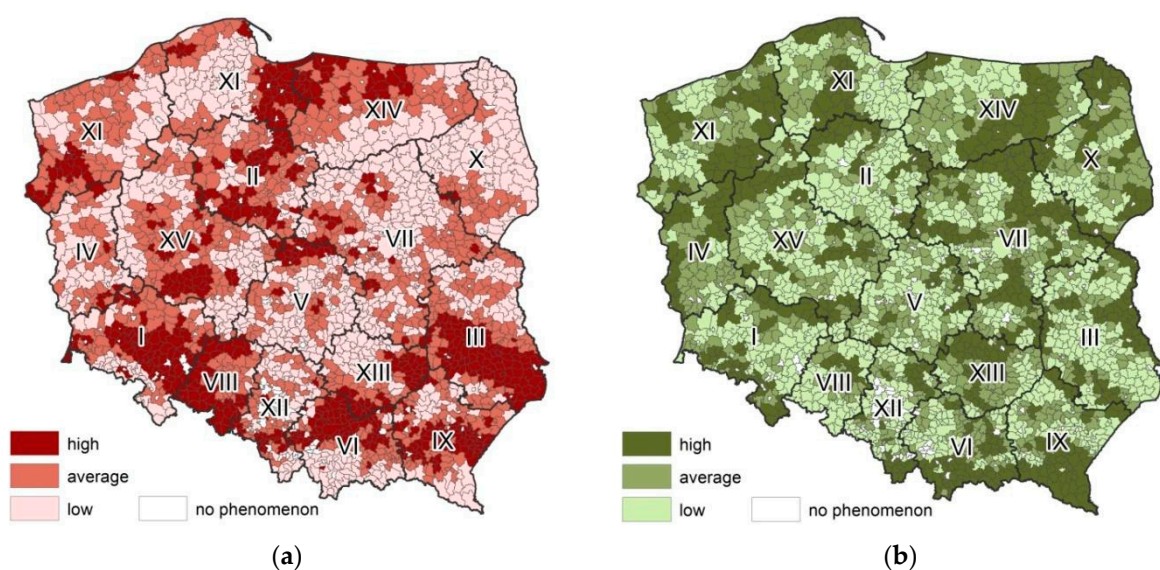

**Figure 4.** The quality of agricultural production space ((**a**); APS) and the natural and ecological quality of areas ((**b**); NEA). Source: own elaboration on [66–68,72,73].

The analysis showed a considerable spatial differentiation of the level of socioeconomic development at the level of communities, confirming two basic rules indicated in numerous studies dealing with the problem under consideration [82–84]. The first rule results from historical conditions and is related to the differences between communities in eastern and western Poland. The second one stems from the differentiation of the level between the centre (agglomerations) and the periphery (rural areas; cf. Figure 3a). The indicated factors, especially the old political and historical divisions, play a significant role in the case of differentiation in the level of agricultural development (cf. Figure 3b) [66]. It is confirmed by the lowest ratings of this sector in Podkarpackie (−0.29) and Małopolskie (−0.24; Austrian partition). The low rating of Śląskie (−0.36) is associated with the domination of the industry (Upper Silesian Industrial District; cf. Figure 3b).

The natural conditions (APS and NEA) refer primarily to the differentiation of soils, which in turn is the result of soil-forming processes dependent on bedrock, climate, water, living organisms, including humans, as well as topography and the passage of time. The quality and usefulness of soils is the most crucial component in the valorisation of the quality of agricultural production space (cf. Figure 4a) [85–87]. Soil production capacity is later reflected in the land use; therefore, high scores of the APS index inversely correlate with NEA (cf. Figure 4b). The high value of the latter should be associated with the areas which show an above-average share of forests, grasslands, water bodies, and legally protected areas in the total area.

### 3.4. Spatial Assessment of the Use of Proenvironmental RDP Funds in the Context of the Conditions of Sustainable Agricultural Development

The conducted spatial analysis allows concluding that proenvironmental activities are more willingly implemented in peripheral (border) regions, mainly in northern, as well as western and eastern parts of Poland. These are the provinces with lower urbanisation levels than the more centrally located ones, more often characterised by high natural and ecological values and proenvironmental forms of land use (NEA = 0.44; see Table 4). In terms of the land use structure, significant areas are covered by forests (Lubuskie) or grasslands (Podlaskie). In some of these areas, more extensive forms of farming are preferred. From the point of view of enhancing natural ecosystems and biodiversity, these are, therefore, preferable areas for such projects.

**Table 4.** Conditions for sustainable agricultural development against the level of use of proenvironmental RDP funds. Source: own elaboration.

| Level of IUF-RDP | Conditions | | | |
|---|---|---|---|---|
| | **LSED** | **LAD** | **APS** | **NEA** |
| Low | −0.62 | −0.07 | 0.08 | −0.43 |
| Average | −0.76 | 0.05 | −0.06 | −0.13 |
| High | −0.62 | 0.32 | −0.51 | 0.44 |

The directions of spatial differentiation of the implemented applications from the AECM and OFS also refer to the areas covered by the NATURA 2000 program. These spatially oriented activities of the beneficiaries help to maintain a balance between environmental resources and the requirements of the economy [7]. Thus, this trend is a strong element of the concept of sustainable development [88] and brings benefits to the natural environment and society. Moreover, the high level of use of proenvironmental RDP measures should be associated with the areas of low natural suitability for agricultural production (APS = −0.51) as well as with a relatively well-developed agricultural sector (LAD = 0.32; see Table 4). On the other hand, the areas with low activity of farmers in introducing proenvironmental management methods show a relatively low average level of agriculture (LAD = −0.07) in the conditions of above-average natural predispositions for agricultural production (APS = 0.08) and very low natural and ecological quality (NEA = −0.43).

Such a territorially oriented implementation of activities shows that the use of biodiversity in agriculture is pragmatic [89], which increases the possibilities of sustainable use and good management of environmental resources [76]. On the other hand, it also contributes to strengthening the environmental ecosystem, which will help preserve biodiversity at the farm level and beyond [90]. Appropriate (i.e., in line with environmental, ecological and agricultural preferences, as well as economic opportunities) territorially oriented activities of AECM and OFS should significantly strengthen the desired economic profile, i.e., aimed at bioeconomy. However, it should be noted that the implementation of proecological activities by farmers is not only motivated by care for the environment. As such, no other behaviours are primarily due to the economic factor related to the possibility of obtaining subsidies from the CAP funds. It is evidenced by the fact that large farms located in the northern and western part of the country (Zachodniopomorskie, Warmińsko-Mazurskie, Lubuskie) benefit to a large extent from the activities of the AECM and OFS in Poland. Thus, this indicates the opposite relationship than, for example, in neighbouring Germany, where smaller farms, focused on more extensive forms of management, are more eager to use funds from proecological activities [39,81].

In the context of bioeconomy development, the economic factor is of crucial importance. One of the three pillars of the European Bioeconomy Strategy is the support of EU funds for proenvironmental forms of farming. The results of the analyses of the authors so far [63,73,77] indicate that it is an important, if not the essential motive that guides Polish farmers in implementing proecological solutions on their farms. The financial factor is also of fundamental importance from the point of view of influencing the broadly understood socioeconomic development of rural areas, mainly since it includes the inflow of funds supporting local economies. From this point of view, funds aimed at the development of proecological activities had an unusually high rank in the communities of Trzcianne—EUR 7.3 million (data for 2015–2019) and Szczecinek—EUR 6.7 million. In both communities, the level of payments was higher than in all communities of Śląskie taken together (a total of EUR 6.5 million). It proves the scale of the inflow of EU funds and the importance of proecological activities, thus supporting the economic development of these communities towards the broadly understood bioeconomy.To better illustrate the rank of this group of funds the ratio (Wr) of the value of proenvironmental EU funds to the size of community budgets (EU payments as percentage of the community budget in 2015–2019) was used. Such targeted analysis confirmed the importance of AECM and OFS measures and their impact on the economic situation of communities. In the Community of Trzcianne,

the ratio as mentioned above was Wr = 0.31 (i.e., the amount of EU payments corresponds to 31% of the community's budget). These funds are of a slightly lower rank in Szczecinek community (ratio IR = 0.15), but their importance for the economic situation of the community is still significant. The communities mentioned above are not the only examples of high positioning of proenvironmental activities in the local economy. In four more administrative units of this level, funds directed at proecological forms of agricultural activity exceeded the level of EUR 5 million: Gołdap (EUR 6.2 million, IR = 0.06), Słońsk (EUR 5.9 million, IR = 0.23), Komańcza (EUR 5.8 million, IR = 0.29), and Drawsko Pomorskie (EUR 5.4 million, IR = 0.06). In total, the value of subsidies in 276 communities exceeded EUR 1 million. On the other hand, in 924 communities (37%) the level of payments did not exceed EUR 100,000. From the economic and bioeconomic development perspective, these funds are of crucial importance in 41 communities (IR > 0.10), and in another 134 they play an essential role (IR 0.05 < 0.10). It can be said that those communities mentioned above are avant-garde in the implementation of proecological solutions strengthening the biodiversity of the natural environment and thus directing development towards the bioeconomy. Nevertheless, the majority of beneficiaries still do not see the potential of ecological solutions that can stimulate the development of local economies. It is confirmed by the fact that in 65% of communities the ratio IR < 0.010 (excluding 191 communities where AECM and OFS activities were not implemented).

The indicator of the use of proenvironmental RDP funds is distinguished by a significant spatial differentiation resulting from the impact of several conditions. The spatial distribution of the indicator was compared to the indicators (expressed in the form of the average normalised value) illustrating the general level of socioeconomic development, level of agriculture, quality indicator of agricultural production space and land use, both at the community level (land with significant natural and ecological values as a percentage of the total community area) and a farm (share of proenvironmental forms of land use in the total area of farms).

The analysis of the correlation of the AECM and OFS utilisation rates showed a relatively weak correlation (−0.5 to +0.5) concerning the adopted indicators (see Table 5). The highest value of the correlation was recorded concerning the index (land with significant natural and ecological values − $r$ = 0.282), which proves a low environmental pressure and concerning the index (agricultural development level (0.233). It proves that the interest among farmers in the implementation and use of funds from agri-environment-climate measures is more significant in areas of natural value and those with a high level of agricultural development. Low values of the correlation were recorded concerning the index (level of socioeconomic development − $r$ = 0.046).

**Table 5.** Dependencies between selected conditions for the development of sustainable agriculture and the level of use of proenvironmental RDP funds. Source: own elaboration.

| Specification | LSED | LAD | APS | NEA | IUF-RDP |
|---|---|---|---|---|---|
| LSED | 1 | 0.105 | −0.025 | −0.078 | 0.046 |
| LAD | 0.105 | 1 | 0.309 | −0.172 | 0.233 |
| APS | −0.025 | 0.309 | 1 | −0.514 | −0.203 |
| NEA | −0.078 | −0.172 | −0.514 | 1 | 0.282 |
| IUF-RDP | 0.046 | 0.233 | −0.203 | 0.282 | 1 |

Nonsignificant correlations are marked in grey ($\alpha$ = 0.05).

Therefore, the above results confirm the fact that the socioeconomic conditions do not have such an enormous impact on the territorial orientation of the implementation of proecological activities as environmental factors. In the context of activities aimed at directing the development of local and regional economies towards the bioeconomy profile, this is of great importance. It clearly shows that the territorialisation of AECM and OFS actions should be related to local conditions (mainly environmental). Then, profiling for the bioeconomy has a chance for sustainable development, which will be sustained by grassroots activities of farmers with the support of the CAP funds. More urbanised,

industrialised regions with a higher level of socioeconomic development base their economic profile more on other industries (e.g., automotive, electromechanical, IT sector, highly specialised services).

The results of the correlation analysis and the distribution of the values of synthetic indicators (see Tables 4 and 5) show that particular attention should be paid to the regional component, as each province shows its specificity, potential and conditions. Thus, it creates different possibilities from the point of view of bioeconomy development. In order to effectively manage and influence the rationality of spending funds from AECM and OFS activities, in line with the objectives of the EU environmental policy (including enhancing biodiversity and bioeconomy), the funds should include a regional component, as is the case in Germany, where each state has specific autonomy in the field of creating a development policy taking into account the existing conditions [91].

Generally, it should be noted that in the whole EU the rank of proecologically oriented activities is gradually increasing, which is a derivative of the change of priorities and the successive strengthening of this direction of development. As a result, in the EU countries, over the last three decades, public funds allocated to the development of organic farming have been gradually increasing and becoming more available [92]. Despite the change of direction in the ecological policy strengthening the bioeconomy, there is still a large gap between funds aimed at conventional agriculture and expenditure on agri-environmental measures, including organic farming (they accounted for about 7%, i.e., nearly EUR 20 billion, of total EU funding for the CAP 2014–2020; European Commission, 2013). Even in the countries with the highest input rates for organic farming in the EU (Germany), this represents only a small part of the total expenditure on agricultural policy [93,94].

## 4. Conclusions

The aim of the study was to identify the spatial level of the use of proenvironmental CAP funds, which constitute an important element in the sustainable development of agriculture, and thus the bioeconomy. Analyses carried out in high spatial resolution allowed showing areas where the proenvironmental management system is widely accepted by farmers and applied on the majority of farmland, thus significantly affecting farm income. It was natural to try to look for factors influencing the differentiation of the studied phenomenon. The detected relationships indicate the complexity of the problem, with stronger relationships related to environmental determinants.

Considering strengthening natural ecosystems and biodiversity, such targeted territorialisation of activities has a solid foundation. It is conducive not only to maintaining the balance between environmental resources but also meeting economic needs, and thus building the foundations for the development of the bioeconomy.

However, the adopted set of conditions does not sufficiently explain the complexity of the process of absorption of proenvironmental RDP funds. Therefore, further research explaining the mechanisms of sustainable agriculture development is essential.

In future activities, it is recommended to strengthen the territorialisation of proecological activities (AECM and OFS, but also others), which should be strongly related to local conditions, mostly natural and ecological. It will provide a stable foundation for sustainable development kept and reinforced by bottom-up activities. The cyclical approach, in line with the current fashions, but not based on local resources, does not create opportunities for the stable long-term development of bioeconomy.

Considering Polish conditions and economic profile, i.e., a significant share of the agricultural sector, from the point of view of bioeconomy, they predispose to the development based on the agricultural and natural potential. It applies primarily to development towards bioenergy, organic food production, and ensuring food security, as well as sustainable agricultural production, which combines production, economic, social, and ethical priorities with environmental security.

The empirical nature of the article contributed to the development of a cognitive thread regarding the impact of EU funds on the development of proenvironmental forms of agricultural management as an element of bioeconomy. The conducted research is also of great application value. It enriches works in the field of bioeconomy, spatial planning, and strategic agriculture with knowledge about the impact of the CAP instruments on the multifunctional and sustainable development of agriculture.

**Author Contributions:** Conceptualisation, A.J.-T., R.R., Ł.W. and M.G.-G. methodology, Ł.W. and R.R. validation, A.J.-T., Ł.W., R.R. and M.B.; formal analysis, A.J.-T., Ł.W., R.R., M.G.-G. and M.B.; investigation, A.J.-T., Ł.W., R.R. and M.B.; resources, Ł.W. and R.R.; data curation, Ł.W. and R.R.; writing—original draft preparation, A.J.-T., R.R., Ł.W. and M.B.; writing—review and editing, A.J.-T., Ł.W., R.R. and M.G.-G.; visualisation, Ł.W.; supervision, A.J.-T., R.R., Ł.W. and M.B.; funding acquisition, R.R. and M.B. All authors have read and agreed to the published version of the manuscript.

**Funding:** This paper was written in the framework of the research project of the National Science Centre (no. DEC-2012/07/B/HS4/00364).

**Conflicts of Interest:** The authors declare no conflict of interest.

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
