# Peer review of "The Agri-Environment-Climate Measure as an Element of the Bioeconomy in Poland—A Spatial Study"

_agriculture, doi:10.3390/agriculture11020110_

Round 1

Reviewer 1 Report

Dear Authors, please find my 62 comments within the attached annotated PDF version of the manuscript. You may respond to the comments directly within the PDF, and attach that PDF when uploading the revised manuscript (if you are going to revise it). Kind regards.

Author Response

Manuscript ID: Agriculture-105407
Title: The agri-environment-climate measure as an element of the bioeconomy in Poland - a spatial study

IN RESPONSE TO REVIEWER 1          
Reviewer 1

The authors would like to thank the Reviewer 1 for such a comprehensive review and constructive comments. The paper has been revised, and all comments have been accounted for. Following the comments and suggestions of the Reviewers, we have made several constructive and substantive changes to our article. Their introduction significantly enriched the substantive and application part of the study and made it more understandable for the reader. Following these, we thoroughly restructured our work. Based on the +literature on the subject, we enriched the introductory part with a "clearer statement about the meaning and justification" of our article, emphasising its substantive and application aspect. An essential element of the changes was:

- authors

According to this comment abstract has been rewritten and novel aspects has been emphasised (lines 40-43, 54-56, 134-141, 257-260, 368-369, 409-411, 421-422)

- Reviewer

-Based on these general findings, what are potential conclusions on how to "develop and influence the innovation policy in the EU model of bioeconomy"? The results here are so broad that one could have easily assumed that these conditions are of importance. Aren't these covered in the current EU model of bioeconomy. What shortages are found here? Please describe both the conclusions and thus the importance of the findings of this study more clear.

- Authors

According to this comment abstract has been rewritten and novel aspects has been emphasised (lines 34-36, 571-578)

Reviewer

“I strongly suggest that more background information about the content of the figure is given, for example scientific approach of what, which project, which study, what do the many abbreviations mean and how must the figure be understood? Is it a schematic diagram, are they milestones or what? Figure descriptions need to be self-explanatory. This needs to be improved.”

„Subsection 2.2 is too short. Integrate it with the next sub-section”.

Authors

In order to fully understand the research scheme in Figure 1. Research procedure, a paragraph was added which refers to the individual test steps, methods, data sources and indicators (abbreviations are referenced). This makes the drawing much more understandable.

With the changes to 2.2. Methods, the subsection has become more extensive, which the authors believe may remain separate. This is important because the subsequent subsections of chapter 2.2 reflect the individual stages of the adopted test procedure.

According to this comment abstract has been rewritten and novel aspects has been emphasised (lines 202-210) ,

Reviewer

“All this reads like a repetition from the introduction to me. What makes all this be a result, when it was already part of the introduction. This doesn't make any sense and should therefore be deleted in my opinion”.

Authors

As suggested by the reviewer, this part of the article was rewritten. Repeating fragments were removed, while the remaining parts, according to the authors, are necessary to understand the essence and importance of agri-environment-climate measures for Polish agriculture.

According to this comment abstract has been rewritten and novel aspects has been emphasised (lines 267-317)

Reviewer 2 Report

The present study assesses the spatial diversification of the level and structure of spending funds for two Rural Development Program measures between 2014 and 2019. It is a strong and topical contribution with a thorough analytical and institutional analysis.

The research question is very original and interesting. The authors successfully embed Polish agricultural policies into the EU Common Agricultural Policy (CAP). These are 1. angri-environment-climate measures (AECM) and 2. organic farming scheme (OFS) aimed at supporting pro-environmental forms of agricultural management in the context of bioeconomy development. Both Voivodeships (reg ional)and local level are differentiated. The study adopts a synthetic indicator of AECM/OFS proposed by the authors. The study consists of three stages: 1. Proposition of and indicator for the use of pro-environmental CAP funds; 2. Assessment palnes - determinants of sustainavle bioagriculture, and 3. Assessment of IUF-RDP: indicator for the use of pro-environmental RDP.

To account for the local governance structures for economic development refer cursorily to the role of subsidiarity and its realization chances, i.e. feasibility, in the context of CAP. Refer to Sadik-Zada, Ferrari & Loewenstein et al. (2018) on the role of federalism and municipalities in the success of commodification of power sector in Latin America.

Show that in contrast to the core EU-15 countries, Poland's agriculture has a legacy of socialism in terms of specialization. This makes this case study especially interesting. This contributes to the justification of the case study.

The paper relies mostly on the bivariate association between the variables and employs correlation analysis. The author must make clear, why the analysis is mostly predicated on correlation analysis, which is an indicator for the association between the variables and not regression analysis. To this end, refer shortly without delving into this econometric issue to Sadik-Zada & Loewenstein (2020), Niklas et al. (2020) and Sadik-Zada & Gatto (2020), where the authors delve into the issue of omitted variable bias.

  • Subsection 2.2 is too short. Integrate it with the next sub-section.

Author Response

Manuscript ID: Agriculture-105407
Title: The agri-environment-climate measure as an element of the bioeconomy in Poland - a spatial study

IN RESPONSE TO REVIEWER 2          
Reviewer 2

IN RESPONSE TO REVIEWER 2

Reviewer 2:

The authors would like to thank the Reviewer 2 for such a comprehensive review and constructive comments. The paper has been revised, and all comments have been accounted for. Following the comments and suggestions of the Reviewers, we have made several constructive and substantive changes to our article. Their introduction significantly enriched the substantive and application part of the study and made it more understandable for the reader. Following these, we thoroughly restructured our work. Based on the +literature on the subject, we enriched the introductory part with a "clearer statement about the meaning and justification" of our article, emphasising its substantive and application aspect. An essential element of the changes was:

- authors

According to this comment abstract has been rewritten and novel aspects has been emphasised (lines 37, 40-43, 54-56, 134-141, 257-260, 368-369, 409-411, 421-422)

Reviewer

The paper relies mostly on the bivariate association between the variables and employs correlation analysis. The author must make clear why the analysis is mostly predicated on correlation analysis, which is an indicator for the association between the variables and not regression analysis. To this end, refer shortly without delving into this econometric issue to Sadik-Zada & Loewenstein (2020), Niklas et al. (2020) and Sadik-Zada & Gatto (2020), where the authors delve into the issue of omitted variable bias.”

authors

The authors are aware of the potential benefits of using regression analysis in this study. Regression analysis allows explaining the value of the dependent variable under the influence of the explanatory variable (or variables). Its capabilities are well confirmed by the works cited by the reviewer [Sadik-Zada & Loewenstein (2020), Niklas et al. (2020), Sadik-Zada & Gatto (2020)].

However, the failure to use regression results from the methodological tests carried out. The model used in the study was built with the use of variables (LSED, LAO, APS, NEA) and explained less than 30% of the variability in the level of the use of the researched CAP funds (IUF RDP). Therefore, the low level of explanation influenced the decision to use other methods, i.e. correlation.

The use of correlation results, inter alia, from the objectives of the study, which was the identification of spatial differentiation in the use of pro-environmental CAP funds and its assessment in terms of selected determinants of agricultural development in Poland. This assessment was made in two ways, using:

- indicators (see table 5) illustrating the average values of the conditions in individual categories of communities taking into account the level of funds utilisation,

- Pearson's linear correlation coefficient (see Table 6).

However, following the reviewer's suggestions, the authors of the article plan further research on pro-environmental EU funds (extension of conditions; data from other institutions) in which they will certainly use regression analysis.

Round 2

Reviewer 1 Report

Well done!

Reviewer 2 Report

The authors provided an improved version.